# The Effect of Mean Platelet Volume/Platelet Count Ratio on Dipper and Non-Dipper Blood Pressure Status

**DOI:** 10.3390/medicina55110742

**Published:** 2019-11-16

**Authors:** Murat Meric, Serkan Yuksel, Metin Coksevim, Okan Gulel

**Affiliations:** Department of Cardiology, Faculty of Medicine, OndokuzMayis University, 55139 Samsun, Turkey; serkany77@yahoo.com (S.Y.); metincoksevim@gmail.com (M.C.);

**Keywords:** mean platelet volume, non-dipper hypertension, platelets

## Abstract

*Background and Objectives:* The mean platelet volume (MPV) represents a possible marker of platelet activation. There is an association between the platelet count (PC) and inflammation and platelet reactivity. We assessed the association between the MPV/PC ratio and circadian alterations in blood pressure (BP). *Material and Methods:* One hundred and twenty subjects in total, 80 hypertensive subjects and 40 healthy subjects (controls), were enrolled in the study group. Twenty four hour ambulatory BP monitoring (ABPM) was applied to all subjects. According to ABPM results, the hypertensive subjects were separated into two groups, such as dippers (*n* = 40) and non-dippers (*n* = 40). In all subjects, the collection of venous peripheral blood samples was performed on admission for PC and MPV measurements. *Results*: The two groups exhibited similar clinical baseline characteristics. A significantly higher MPV/PC ratio was determined in non-dippers compared to that in dippers and normotensives. The higher MPV/PC ratio was observed in non-dippers in comparison with that in dippers and normotensives (0.046 ± 0.007 to 0.032 ± 0.004 fL/[10^9^/L]; 0.046 ± 0.007 to 0.026 ± 0.004 fL/[10^9^/L], *p* < 0.001, respectively). A receiver operating characteristic (ROC) curve analysis showed that the optimum cut-off value of the MPV/PC ratio for predicting non-dipping patterns in hypertensive patients was 0.036 (area under the curve [AUC]: 0.98, *p* < 0.001). According to the cut-off value, sensitivity and specificity were found to be 95% and 95%, respectively. *Conclusions:* The higher MPV/PC ratio was determined in non-dipper hypertensive subjects in comparison with that in dipper hypertensive subjects. An elevation of platelet activity and an increase in thrombus burden are reflected by an increase in the MPV/PC ratio. The MPV/PC ratio may underlie the increase in cardiovascular risk in non-dippers compared to that in dippers.

## 1. Introduction

Blood pressure (BP) and heart rate undergo circadian changes during the day. It is possible to utilize ambulatory blood pressure monitoring (ABPM) for detecting circadian rhythms of BP parameters correlated with an adverse cardiovascular prognosis. Among healthy individuals, a nighttime decrease of 10–20% in daytime BP is defined as “dipper,” whereas a decrease of <10% is defined as “non-dipper” [1,2,3]. In prospective studies, the absence of a nighttime fall or elevated nocturnal BP in comparison with daytime BP were independent risk factors for cardiovascular diseases [4,5,6,7].

The mean platelet volume (MPV) represents a possible marker of platelet activation, and an elevated MPV is related to an increase in cardiovascular mortality in subjects with acute coronary syndrome [8]. In previous research, an association was determined between the platelet count (PC) and inflammation and platelet reactivity [9,10]. Ischemic stroke patients and patients with ischemic heart disease had lower PCs in comparison with those of healthy controls [11,12].

The MPV/PC ratio represents a novel marker for cardiovascular risk. Azab et al. [13] indicated that an increased MPV/PC ratio better predicted long-term cardiovascular mortality in subjects having non-ST elevation acute coronary syndrome as compared to MPV and PC alone. Previous studies demonstrated that platelet size and PC were inversely related in healthy individuals, showing that MPV and PC should be considered as ratios rather than independent variables [9,14,15,16].

In this research, it was aimed to assess the relationship between the MPV/PC ratio and circadian changes in BP.

## 2. Material and Methods

### 2.1. Study Population

One hundred and twenty subjects in total, 80 hypertensive subjects and 40 healthy controls (males: *n* = 17, mean age of 51.6 ± 15 years), were included in the study group. Twenty four hour ABPM was applied to all patients. In accordance with the findings of ABPM, the hypertensive subjects were separated into two groups as 40 dippers (males, *n* = 17; mean age of 53.3 ± 14.3 years) and 40 non-dippers (males, *n* = 16; mean age of 53.5 ± 13 years). ABPM results confirmed that the control group patients were normotensive, with dipper profiles.

Hypertension was described as systolic BP (SBP) ≥140 mmHg or diastolic BP (DBP) ≥90 mmHg and/or usage of antihypertensive drug therapy [2]. After hypertension was diagnosed, the participants underwent ABPM.

The criteria for exclusion from the study were as follows: secondary hypertension, cardiac failure, renal or hepatic dysfunction, systemic inflammatory diseases, infectious diseases, stroke, valvular diseases, arrhythmias, drug use that may influence platelet number and function hematological abnormalities, coronary artery diseases and diabetes mellitus.

Clinical BP measurements were performed in the morning by a mercury sphingomanometer, and the average of three measurements was utilized. Prior to the measurements, the individual rested for a minimum of 5 min. Each individual was instructed not to drink tea or coffee for 1 h prior to the test and not to smoke for 30 min prior to the test. During BP measurements, the arm of the patient was supported at heart level, and BP measurement was performed in a sitting position. The measurements were performed on both arms, and the highest value was recorded [2,17].

Clinical and demographic features, such as sex, age, smoking habits, and the use of antihypertensive drugs, were recorded. Recording of fasting blood glucose levels, creatinine levels, and fasting serum lipid status, such as total cholesterol, high-density lipoprotein, low-density lipoprotein, and triglyceride levels, was also performed. Calculation of the body mass index (BMI) was performed using the following formula: weight (kg) divided by height squared (m^2^).

The research was conducted in accordance with the principles of the Declaration of Helsinki. Approval for the study was acquired from the local ethics committee (OMU KAEK 2019/656-26.09.2019), and written informed consent was acquired from all the subjects included in the study.

### 2.2. Ambulatory Blood Pressure Recordings

A Tracker NIBP2 (Del Mar Reynolds Medical Ltd., Hertford, U.K.) oscillometric monitoring device was used for conducting ABPM. Using this device, daytime BP was measured and recorded for 24 h at intervals of 15 min. The measurement of nighttime BP was performed every 30 min at night. The cuff of the ABPM device was placed on the nondominant arm in case the BP difference between the two arms was less than 10 mmHg in the clinical BP measurement and on the arm of the highest measurement in case the difference was higher than 10 mmHg. Care was taken to ensure that the difference between the clinical BP measured at the office and the value measured by the device was not higher than 5 mmHg [2,3,17]. The recordings were analyzed using interactive software.

Information about the procedure was provided to the patients, and they were instructed to perform their daily activities as normal. They were also advised to avoid excessive activity and to keep their arms at heart level during the BP measurements. As the sleep and wakefulness periods of the participants differed, the classification of daytime hours and nighttime hours was determined separately for each individual. The definitions of ‘day’ and ‘night’ periods in our clinics were based on the most common answers to a questionnaire in which patients were asked about their sleeping behavior. Using the hourly averages of ABPM recordings, daytime, nighttime, and 24 h averages of SBP, DBP, and mean BP were computed for every subject. Recordings were accepted if more than 80% of the raw data were valid. Participants with incomplete or invalid ABPM were asked to repeat this procedure 1 week later (*n* = 4). Patients were asked to keep a diary of occupational activities, sleep, and awake time, as well as the time of meals [3,17].

Based on these recordings, individuals in whom a BP decrease of 10% or higher was detected at nighttime were considered to be dipper hypertensives, whereas those with a BP decrease below 10% were considered to be non-dipper hypertensives in accordance with the criterion of Verdecchia et al. [1,2,3,6,17].

### 2.3. Laboratory Testing

Blood samples were collected in the morning after a 12 h fasting period, after a rest for 30 min. In all subjects, the collection of venous peripheral blood samples was performed on admission to measure PC and MPV. Blood sample collection was carried out in standardized tubes that contained dipotassium ethylenedinitrilotetraacetic acid (EDTA) and kept at room temperature (about 23 °C). The blood samples were sent directly to the emergency department laboratory and analyzed immediately as per standard protocol. PC and MPV were analyzed using a fully automated hematological analyzer (Sysmex XN-100; Sysmex Co., Kobe, Japan) within 30 min after blood sampling. In accordance with our laboratory practice, normal MPV values were considered to be 6.1–8.9 fL.

### 2.4. Statistical Analysis

The research data were uploaded to a computer and evaluated using the Statistical Package for Social Sciences for Windows 22.0 (SPSS Inc., Chicago, IL, USA). Descriptive statistics were presented as mean ± standard deviation, frequency distribution, and percentage. Categorical variables were evaluated by Pearson’s chi-square test. Whether the variables complied with a normal distribution was studied by visual (histogram and probability graphs) and analytical methods (the Shapiro–Wilk test). The Kruskal–Wallis test was conducted to determine the statistical significance between the three independent groups for variables found to be inconsistent with a normal distribution. The Bonferroni correction was applied in post-hoc paired comparisons performed to identify the source of the difference when a significant difference was found. The relationship between the variables was assessed by Spearman’s correlation test. The diagnostic capability of the MPV/PC ratio in predicting non-dipper status was evaluated by a ROC curve analysis. The statistical significance level was accepted to be *p* < 0.05.

## 3. Results

The demographic features, clinical and laboratory characteristics, and the drug use of the patient and control groups are presented in Table 1. No significant differences were determined between the groups with regard to age, sex, creatine, BMI, hemoglobin level, lipid profiles, rate of smoking, fasting levels of glucose, white blood cell count, and antihypertensive drug usage of the subjects.

In accordance with expectations, significantly higher clinical BP was found in dipper and non-dipper hypertensive subjects as compared to that of normotensives (SBP: 144.4 ± 15.9 and 148.2 ± 21.8 vs. 126.6 ± 8.2 mmHg, *p* < 0.001; DBP: 83.4 ± 12.3 and 83.7 ± 13.3 vs. 76 ± 8.4 mmHg, *p* = 0.016), whereas it was similar in dippers and non-dippers.

As shown in Table 2, significantly higher nighttime SBP and nighttime average BP were determined in non-dippers in comparison with those in dippers. Similarly, daytime average BP, 24 h SBP, and 24 h average BP values were found to be higher in non-dippers.

A significantly higher MPV/PC ratio was determined in non-dippers in comparison with that in dippers and normotensives. As is observed in Table 1 and Figure 1, the MPV/PC ratio in non-dippers was higher than that in dippers and normotensives (0.046 ± 0.007 to 0.032 ± 0.004 fL/(10^9^/L); 0.046 ± 0.007 to 0.026 ± 0.004 fL/(10^9^/L), *p* < 0.001, respectively).

In hypertensive patients, ROC curves revealed a correlation between the non-dipping status and MPV/PC ratio. The ROC analysis demonstrated that the optimum cut-off value for the MPV/PC ratio for predicting non-dipping patterns in hypertensive groups was 0.036 (area under the curve [AUC]: 0.98, *p* < 0.001). According to the cut-off value, sensitivity and specificity were found to be 95% and 95%, respectively. 

A significant difference was found in the usage of antihypertensive drugs among dipper and non-dipper patients. No drug use was recorded among normotensives (Table 1).

## 4. Discussion

Endothelial damage and dysfunction, insulin resistance, inflammatory and platelet activation, and coagulation abnormalities contribute to the pathophysiology of hypertension, resulting in an increase in the risk of a prothrombotic state [18,19]. It is a known fact that non-dipper hypertensive subjects have an increased risk of endothelial dysfunction and inflammation, with a subsequent risk of target organ damage [1,2,3,4,5,6,7]. In this research, we compared MPV/PC levels in a group of non-dipper hypertensive subjects with those in a group of dipper hypertensive subjects. The results revealed the higher MPV/PC ratio in non-dipper hypertensive subjects in comparison with that in dipper hypertensive subjects.

BP measurement is the most significant step in the diagnosis and treatment of hypertension. The BP values obtained in the office reflect only the measurements at that moment in time and do not provide information on 24 h BP. BP exhibits diurnal changes. Due to decreased sympathetic activity during nocturnal sleep and increased vagal tone, BP decreases significantly and increases rapidly upon morning waking. The definitions of “dipper” and “non-dipper” are based on the prognostic importance of decreased BP at night [1,2,3,4,5,6]. Previous studies reported that non-dipper hypertension was an independent risk factor for cardiovascular diseases [1,2,3,4,5,6]. Moreover, they showed that target organ damage was much more common in non-dipper hypertensive patients than in dippers [1,2,3,4,5,6]. Furthermore, autonomic dysfunction, heart rate variability disorders, glucose metabolism disorders, and vascular complications were more frequent in non-dipper hypertensive patients, together with left ventricular hypertrophy, cerebrovascular events, microalbuminuria, cardiovascular mortality and morbidity [1,2,3,4,5,6]. In predicting cardiovascular events, 24 h BP values recorded by ABPM seemed to be superior to office values, and nighttime values appeared to be superior to daytime values [1,2,3,4,5,6]. The worst prognosis for cardiovascular outcomes was observed in reverse dipper and non-dipper patients, respectively. According to a previous study, BP values over a 24 h period should be taken into account in antihypertensive treatment plans [20].

Some previous studies revealed an association between MPV and hypertension in various groups of patients. Nadar et al. [19] revealed that hypertensive subjects with organ damage, such as previous myocardial infarctions, strokes, microalbuminuria/proteinuria, angina, and left ventricular hypertrophy had elevated MPV levels as compared to those in hypertensive subjects without target organ damage. Other studies revealed an association between MPV values and dipper and non-dipper hypertension types [21,22]. In both studies, increased MPV values were determined in non-dipper patients in comparison with those in dipper and normotensive patients.

The MPV represents a marker of platelet size and activation, and an increase in its level indicates activation of large platelets [21,22,23,24]. Various clinical studies showed that activation of large platelets involved denser granules that were more metabolically and enzymatically active and had greater thrombotic potential in comparison with small platelets [24]. In other studies, an increase in the MPV was related to mortality rates following acute myocardial infarction and restenosis [25]. Chandrashekar et al. [26] indicated that increased platelet activation could underlie the association between inflammation and atherosclerotic plaque development.

A number of previous studies revealed a relationship between increased MPV and increased cardiovascular events [21,27,28]. At present, most methods available for the analysis of platelet activity are expensive and require a lot of time and specific tests and instruments [29]. Unlike these methods, the MPV can be measured easily and cheaply. A high MPV demonstrates the presence of larger and more reactive platelets. More prothrombotic material is released by larger and reactive platelets, which accelerates thrombus formation and increases the risk of many diseases [30].

Previous research reported chronic rather than acute alterations in the MPV, suggesting that these alterations were associated with chronic diseases, such as hypertension [31]. Some studies also indicated that the MPV was associated with hypertension. For example, Varol et al. [32] revealed higher MPV values in hypertensive individuals in comparison with prehypertensives and higher MPV values in prehypertensives in comparison with normotensive individuals. MPV values were also reported to be elevated in subjects with resistant hypertension as compared to those in cases of controlled hypertension or normotension [28]. The above-mentioned studies indicated that MPV values were positively associated with BP. Nevertheless, these studies did not demonstrate that the MPV could independently predict hypertension. 

It was shown that non-dipper hypertensive patients are more susceptible to develop vascular damage in comparison with dippers [28]. The non-dipping pattern is associated with high levels of molecules related to endothelial dysfunction, platelet activation, altered haemostasis and atherosclerosis, such as von Willebrand factor, soluble intercellular adhesion molecule-1, soluble CD40 ligand, D-dimer and plasminogen activator inhibitor-1, all of which contribute to the link between non-dipping pattern and, atherosclerosis and cardiovascular disease [33]. In addition, thrombocyte activation observed in non-dipper hypertensive patients resulting from increased oxidation and shear stress can be a possible mechanism associating non-dipper pattern with increased MPV [28]. Non-dipper hypertensive patients show impaired autonomic system functions including abnormal parasympathetic and increased sympathetic nervous system activity [34]. Sürgit et al. [28] suggested that overactivated sympathetic nervous system could cause shape changes in thrombocyte activation through the stimulation of alpha-2 adrenoreceptors. They also suggested that larger activated platelets sequestered in the spleen, following increased adrenaline levels contributing to the increased MPV levels, can be released to the circulatory system, which, in turn, could be a mechanism promoting increased MPV levels in non-dipper hypertensive patients [28].

Azab et al. [13] indicated that a comparatively high MPV and a low PC reflected increased platelet reactivity and aggregation. They demonstrated the advantage of the MPV/PC ratio over the MPV alone in determining the prognosis in non-ST-elevation myocardial infarctions. Guenancia et al. [23] demonstrated that an increased MPV/PC ratio was related to ischemic strokes after acute myocardial infarctions.

Previous studies indicated that the MPV and PC were inversely related in healthy populations [9,14,15]. Therefore, MPV and PCs should be interpreted as ratios rather than independent variables. Quan et al. [16] referred to the MPV/PC ratio as the platelet ratio and showed that the platelet ratio represented an independent predictor of 90-day outcomes in stroke subjects having large artery atherosclerosis. Similarly, the MPV/PC ratio may be a cardiovascular risk marker in non-dipper hypertensive subjects.

Numerous studies have reported that non-dippers are of a higher risk group for atherosclerotic events and for high incidence of target organ damage in comparison with dippers [1,2,3,4,5,6,7]. It is well-documented that increased platelet activation is crucial in the development of atherosclerosis [35]. It was reported that increased platelet activity was associated with increased platelet volüme [21,22,23,24]. MPV is a determinant factor in platelet activation [21,22,23,24]. Many studies have shown that the increased MPV/PC ratio is an indicator of increased platelet activation and aggregation [7,13,15,16,23]. Further investigations are required to clarify whether calculation of MPV/PC ratio could be used as a marker for predicting cardiovascular risk in this setting.

Table 1 presents the percentage of patients using diuretics as hypertensive drugs. The diuretic used is of the thiazide group and it was used in low doses at fixed doses in combination with angiotensin-converting-enzyme inhibitor and angiotensin receptor blocker. A significant correlation between the use of statins and antihypertensive drugs and, MPV has been reported in previous studies [36]. The fact that MPV can be affected by various factors suggests that MPV/PC ratio should be taken into account rather than MPV alone. In our study, we aimed to investigate the correlation between MPV/PC ratio and circadian change in blood pressure, to evaluate its estimated value and to reveal that it can be an indicator for dysfunction of thrombocyte activation in non-dipper hypertensive patients. In previous studies, a significantly higher MPV was found in hypertensive subjects in comparison with that in normotensive subjects [19], and platelet dysfunction in hypertensive subjects was put forward as a possible reason for an increase in cardiovascular mortality [37]. In patients with hypertension, a high risk of cardiovascular diseases was determined in the non-dipping group. There may be an association between this additive risk and an increase in platelet activation in non-dippers.

## 5. Conclusions

The higher MPV/PC ratio was found in non-dipper hypertensive subjects in comparison with that in dipper hypertensive subjects. An increase in the MPV/PC ratio indicates an elevation of platelet activity and an increase in thrombus burden. The MPV/PC ratio may underlie the increased cardiovascular risk in non-dippers compared to that in dippers.

## Figures and Tables

**Figure 1 medicina-55-00742-f001:**
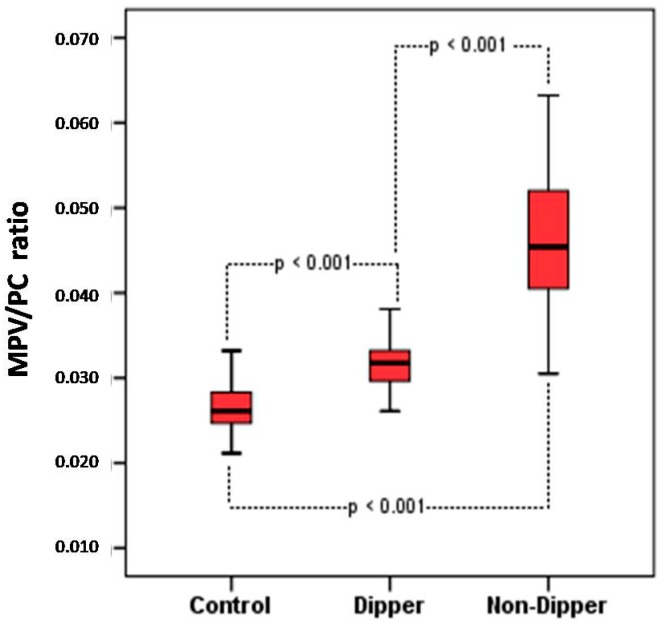
Comparison of the MPV/PC ratio in non-dippers as compared with that in dippers and controls. MPV, mean platelet volume; PC, platelet count.

**Table 1 medicina-55-00742-t001:** Demographic, clinical, and laboratory characteristics of the study population.

	Control (*n* = 40)	Dipper (*n* = 40)	Non-Dipper (*n* = 40)	*p* Value
Age (y), mean ± SD	51.6 ± 15.0	53.3 ± 14.3	53.5 ± 13.0	0.774
Men, *n* (%)	17 (42.5)	17 (42.5)	16 (40.0)	0.966
BMI (kg/m^2^), mean ± SD	28.6 ± 4.2	28.2 ± 4.3	28.2 ± 4.2	0.830
Smokers, *n* (%)	10 (33.3)	13 (32.5)	12 (30.0)	0.951
Clinical SBP (mmHg), mean ± SD	126.6 ± 8.2 ^bc^	144.4 ± 15.9	148.2 ± 21.8	<0.001
Clinical DBP (mmHg), mean ± SD	76.0 ± 8.4 ^bc^	83.4 ± 12.3	83.7 ± 13.3	0.016
Total cholesterol (mg/dL), mean ± SD	202.6 ± 38.1 ^b^	173.0 ± 64.5	193.1 ± 41.0	0.044
Low-density lipoprotein (mg/dL), mean ± SD	124.7 ± 34.1	118.7 ± 38.8	112.9 ± 32.4	0.377
High-density lipoprotein (mg/dL), mean ± SD	49.2 ± 10.8	44.9 ± 11.2	47.7 ± 12.9	0.270
Triglycerides (mg/dL), mean ± SD	150.9 ± 64.5	144.5 ± 64.0	132.2 ± 64.3	0.271
Creatinine (mg/dL), mean ± SD	0.84 ± 0.18	0.89 ± 0.26	0.81 ± 0.24	0.134
Fasting glucose (mg/dL), mean ± SD	102.1 ± 12.1	100.4 ± 15.5	99.2 ± 10.3	0.563
Hemoglobin (g/dL), mean ± SD	14.2 ± 1.4	13.7 ± 1.3	13.7 ± 1.4	0.182
WBC (10^3^/mm^3^), mean ± SD	7.5 ± 1.5	7.5 ± 1.5	7.3 ± 1.8	0.857
MPV (fL), mean ± SD	8.0 ± 0.4 ^bc^	8.4 ± 0.8 ^c^	9.7 ± 1.4	<0.001
PC, mean ± SD	306.2 ± 40.2 ^bc^	267.7 ± 26.1 ^c^	211.8 ± 27.9	<0.001
MPV/PC ratio (fL/(10^9^/L), mean ± SD	0.026 ± 0.004 ^bc^	0.032 ± 0.004 ^c^	0.046 ± 0.007	<0.001
Medical treatments				
ACE inhibitors, *n* (%)	–	19 (47.5)	16 (40.0)	0.499
ARB, *n* (%)	–	10 (25.0)	14 (35.0)	0.329
Beta-blockers, *n* (%)	–	8 (20.0)	13 (32.5)	0.204
Calcium-channel blockers, *n* (%)	–	10 (25.0)	12 (30.0)	0.617
Diuretics, *n* (%)	–	18 (45.0)	16 (40.0)	0.651

%: Column percent; ^b^: Posthoc pairwise comparisons (control vs. dipper) *p* < 0.018; ^c^: Posthoc pairwise comparisons (control vs. non-dipper) *p* < 0.018.

**Table 2 medicina-55-00742-t002:** Comparison of the 24 h ambulatory blood pressure monitoring results of the dippers and non-dippers.

	Dipper (*n* = 40)	Non-Dipper (*n* = 40)	*p* Value
	Mean ± SD	Mean ± SD	
Daytime SBP (mmHg)	137.4 ± 22.8	136.0 ± 18.2	0.862
Daytime DBP (mmHg)	85.6 ± 21.9	77.6 ± 12.1	0.256
Daytime average BP (mmHg)	83.9 ± 13.5	91.3 ± 12.8	0.027
Nighttime SBP (mmHg)	106.9 ± 17.6	133.6 ± 18.9	<0.001
Nighttime DBP (mmHg)	75.4 ± 21.6	74.5 ± 12.1	0.187
Nighttime average BP (mmHg)	72.9 ± 10.0	88.5 ± 13.3	<0.001
24-h SBP (mmHg)	113.9 ± 25.5	135.7 ± 17.8	<0.001
24-h DBP (mmHg)	83.0 ± 21.2	77.1 ± 11.8	0.547
24-h average BP (mmHg)	81.2 ± 12.0	90.6 ± 12.5	0.002

SBP, systolic blood pressure; DBP, diastolic blood pressure; BP, blood pressure.

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
