# Peer review of "The Effect of Mean Platelet Volume/Platelet Count Ratio on Dipper and Non-Dipper Blood Pressure Status"

_medicina, 2019, doi:10.3390/medicina55110742_

Round 1

Reviewer 1 Report

The work is worthy of being published.

Author Response

Thank you for your kind conclusion.

Reviewer 2 Report

The paper Meric M et al. explore the relationship between non dipper profile, platelet activation and cardiovascular risk. The authors collected data from 120 individuals, 80 hypertensives and 40 normotensives, that were studied by means of demographic, clinical, laboratory data and 24-h BP values. Moreover, they suggested a new marker of platelet activation (MPV/PC ratio), that correlates with cardiovascular diseases, as documented by previous literature, and has the advantage of been easy to detect and cheap. The paper is well written, easy to read, has a correct length and the references cited are appropriate. However, this is a cross sectional analysis and due to the nature of this analysis the authors cannot state a cause-effect relationship between high MCV/PC ratio and cardiovascular events. 

It is well known the correlation between platelet activation and cardiovascular disease, as well as association between non dipping status and increased cardiovascular risk, however the authors did not explain which is the possible underlying mechanism between platelet activation and non dipping profile.

Did the patients take diuretics as anti-hypertensives drugs? As diuretics may influence the haematocrit levels, they should be taken into account.

Page 1, line 35 and Page 3 line 97. The authors correctly identified dippers and non dippers according to whether BP decreases 10% or more in the former group and less than 10 % in the latter group, and reported as reference the paper by O’ Brien et al (ref n 1) in the first case and the paper by Verdecchia and co-workers in the second case (ref n4). However, they should list as references also the last ESH 2018 guidelines (J Hypertens 2018;36:1953-2041) and the position paper on Blood pressure monitoring of the European society of Hypertension (J Hypertens 2013; 31(9):1731-68). The same references should be added at page 5 line 171.

Page 2 Line 66-71, Methods paragraph. In the study BP was correctly detected on both arms, the highest value was chosen, and triplicate measurements were recorded. The authors should add appropriate references supporting their methods: ESH 2018 Guidelines (J Hypertens 2018;36:1953-2041; American Heart Association guidelines for Measurement of Blood Pressure in Humans (Hypertension. 2019;73:e35–e66.).

Page 2-3, line 80-93. The authors did not mention the diary card of the patients performing ABPM. It is essential to correctly interpret the recording of the patient, if patients did not fill it please add this aspect to the limitation paragraph, otherwise add it in the text. In the same paragraph the authors explained the technique followed to program the ABPM device, please added the appropriate references: ESH 2018 Guidelines (J Hypertens 2018;36:1953-2041; American Heart Association guidelines for Measurement of Blood Pressure in Humans (Hypertension. 2019;73:e35–e66.); position paper on Blood pressure monitoring of the European society of Hypertension (J Hypertens 2013; 31(9):1731-68).  Finally, the authors should declare the percentage of valid recordings accepted to consider or refuse the ABPM, usually a 30% of non-valid recordings makes the ABPM non reliable.

The authors stated that the night-time period was determined separately for each individual. How did the authors identified this period, probably by patient’s diary card. Please clarify this aspect in the method paragraph (pag 3 line 93).

Page 3 line 124 correct body mass index with BMI

Pag 3 line118 , please correct MPV/PLT with MPV/PC

Author Response

Dear Reviewer,

I have completed the corrections and revisions suggested by you for the manuscript entitled " the effect of mean platelet volume/plateret count ratio on dipper and non-dipper blood pressure status"

Sincerely yours,

My answer to your comments is as follows

Point 1. "It is well known the correlation between platelet activation and cardiovascular disease, as well as association between non dipping status and increased cardiovascular risk, however the authors did not explain which is the possible underlying mechanism between platelet activation and non dipping profile"

Many studies have shown that non-dippers carry a high risk of atherosclerotic events (myocardial or cerebral infarction) and also high incidence of target organ damage compared with dippers. As known, increased platelet activation plays an important role in the development of atherosclerosis. Increased platelet activity is associated with increased platelet volume. Mean platelet volume (MPV) is a determinant of platelet activation. Many studies have shown that the increased MPV/PC ratio is an indicator of increased platelet activation and aggregation. Further investigations are required to clarify whether calculation of MPV/PC ratio could be used as a marker for predicting in this setting.

In our study, we aimed to investigate the correlation between MPV/PC ratio and circadian change in blood pressure; to evaluate its estimated value and to reveal that it can be an indicator or dysfunction of thrombocyte activation in non-dipper hypertensive patients. Studies have showed that thrombocyte dysfunction in hypertensive patients can be a potential cause of the increase in mortality rate due to cardiovascular reasons. In hypertension patients, the non-dipping group has a high risk of cardiovascular disease. This adaptive risk can be associated with the increase in thrombocyte activation.

Point 2. " Did the patients take diüretics as anti-hypertensives drugs? As diüretics may influence the haematocrit levels, they should be taken into account."

Table 1 presents the percentage of patients using diuretics as anti-hypertensive drugs . The diuretic used is of the thiazide group and it was used in low doses at fixed doses in combination with ARB and ACEI.The diuretic affect the haematocrit levels but, to the best of our knowledge, there is no reference in literature as to their effect on MPV and platelet.

Point 3. " Page 1, line 35 and Page 3 line 97. The authors correctly identified dippers and nondippers according to whether BP decreased 10% or more in the former group and less than 10 % in the latter group, and reported as reference the paper by O'Brien et al (ref n 1) in the first case and the paper by Verdecchia and co-workers in the second case (ref n4). However, they should list as references also the last ESH 2018 guidelines (J Hypertens 2018;36:1953-2041) and the position paper on Blood pressure monitoring of the European society of Hypertension (J Hypertens 2013;31(9):1731-68). The same references should be added at page 5 line 171."

References (reference 2 and 3) have been added to the article.Reference numbers have been changed.

Point 4." Page 2 Line 66-71, Methods paragraphs. In the study BP was correctly detected on both arms, the highest value was chosen, and triplicate measurements were recorded. The authors should add appropriate references supporting their methods:ESC 2018 Guidelines (J Hypertens 2018;36:1953-2041; American Heart Association guidelines for Measurement of Blood Pressure in Humans (Hypertension. 2019;73:e35-e66)."

References have been added to the article accordingly as required.

Point 5." Page 2-3, line 80-93. The authors did not mention the diary card of the patients performing ABPM. It is essential to correctly interpret the recording of the patient, if patients did not fill it please  add this aspect to the limitation paragraph, otherwise add it in the text. In the same paragraph the authors explained the technique followed the program the APBM device, please added the appropriate references: ESC 2018 Guidelines (J Hypertens 2018:36:1953-2041); American Heart Association guidelines for Measurement of Blood Pressure in Humans (Hypertension, 2019;73:e35-e66); position paper on Blood pressure monitoring of the European society of Hypertension (J Hypertens 2013; 31(9):1731-68). Finally, the authors should declare the percentage of valid recordings accepted to consider or refuse the ABPM, usually a 30% of non-valid recordings makes the ABPM non reliable."

References have been added to the article (ref 2,3 and 17). The required amendments have been made in the material and methods section of the article.

Point 6."The authors stated that the night-time period was determined separately for each individual. How did the authors identified this period, probably by patient's diary card. Please clarify this aspect in the method paragraph (pag 3 line 93).

Explained in the material and method section

Point 7. Page 3 line 127 correct body mass index with BMI, Pag 3 line 118, please correct MPV/PLT with MPV/PC"

Page 3 line 124,  corrected as BMI.

Page 3 line 118, corrected as MPV/PC

Reviewer 3 Report

The manuscript is well drafted and the results are displayed and analyzed well. Overall the manuscript is in good condition to be printed in the journal.

Author Response

Thank you for your kind conclusion

Round 2

Reviewer 2 Report

The authors answered to all my queries. Please added the extensive explanation to my 1st and second queries in the discussion praragraph

Author Response

Response to Reviewer 2 Comments

Point 1: The authors answered to all my queries. Please added the extensive explanation to my 1st and second queries in the discussion paragraph.

Response 1: Corrections suggested by you were added to revised manuscript text in the discussion section.